# Imaging and controlling coherent phonon wave packets in single graphene nanoribbons

Yang Luo [1], Alberto Martin-Jimenez[1], Michele Pisarra [2], Fernando Martin [3,4,5], Manish Garg [1] ✉ & Klaus Kern [1,6]

The motion of atoms is at the heart of any chemical or structural transformation in molecules and materials. Upon activation of this motion by an external source, several (usually many) vibrational modes can be coherently coupled, thus facilitating the chemical or structural phase transformation. These coherent dynamics occur on the ultrafast timescale, as revealed, e.g., by nonlocal ultrafast vibrational spectroscopic measurements in bulk molecular ensembles and solids. Tracking and controlling vibrational coherences locally at the atomic and molecular scales is, however, much more challenging and in fact has remained elusive so far. Here, we demonstrate that the vibrational coherences induced by broadband laser pulses on a single graphene nanoribbon (GNR) can be probed by femtosecond coherent anti-Stokes Raman spectroscopy (CARS) when performed in a scanning tunnelling microscope (STM). In addition to determining dephasing (~440 fs) and population decay times (~1.8 ps) of the generated phonon wave packets, we are able to track and control the corresponding quantum coherences, which we show to evolve on time scales as short as ~70 fs. We demonstrate that a two-dimensional frequency correlation spectrum unequivocally reveals the quantum couplings between different phonon modes in the GNR.

The periodic collective motion of chemically bonded atoms around their equilibrium configuration is associated with discrete frequencies that are characteristic of the molecule or material to which these atoms belong. This is the key for chemical sensing and structural determination based on vibrational spectroscopies, as, e.g., Raman spectroscopy. By generating and tracking vibrational wavepackets, coherent Raman scattering (CRS) techniques, such as time-resolved coherent anti-Stokes Raman spectroscopy (CARS) and impulsively stimulated Raman spectroscopy (ISRS), can track the dynamics of vibrational motion in bulk molecular ensembles, and thereby unravel isomerization, charge-transfer and conical intersection dynamics, with

femtosecond time resolution[1–10]. Moreover, coherently manipulating phonon modes has a unique potential to control the electronic phases of materials. For example, selective excitation of specific phonon modes has been successfully used to drive complex solids into metastable states with novel physical properties[11,12]. Owing to the nonlocal nature of the ultrafast vibrational excitation, however, the degree of control achieved so far is limited. In femtochemistry, due to the low absorption cross-section of molecules to optical excitation, the success to drive chemical transformations by selective excitation of vibrational coherences is restricted to molecular ensembles in gas and liquid phases[13]. In quantum materials, on the other hand, the direct

[1]Max Planck Institute for Solid State Research, Heisenbergstr. 1, 70569 Stuttgart, Germany. [2]INFN-LNF, Gruppo Collegato di Cosenza, Via P. Bucci, cubo 31C, 87036 Rende (CS), Italy. [3]Instituto Madrileño de Estudios Avanzados en Nanociencia (IMDEA Nano), Faraday 9, Cantoblanco, 28049 Madrid, Spain. [4]Departamento de Química, Módulo 13, Universidad Autónoma de Madrid, 28049 Madrid, Spain. [5]Condensed Matter Physics Center (IFIMAC), Universidad Autónoma de Madrid, 28049 Madrid, Spain. [6]Institut de Physique, Ecole Polytechnique Fédérale de Lausanne, 1015 Lausanne, Switzerland. ✉e-mail: mgarg@fkf.mpg.de

correlation between optical excitation and changes in the electronic properties has proven to be cumbersome.

An important advancement to overcome these limitations is the development of a local probe, allowing for exciting and probing the selective vibrational modes and their dynamics simultaneously in space and time. Ideally, this probe should also be able to capture the electronic signature of the molecule/material under investigation. The capability to effectively excite and probe atomic vibrations at the molecular scale can be realised by enhancing the light-matter inter-action with localised surface plasmons[14]. In particular, by spatially localising the incoming light below the apex of a plasmonic nanotip, tip-enhanced Raman spectroscopy (TERS) has enabled the nanoscale chemical analysis of molecules directly in real-space[15–34], i.e., with atomic resolution. The spatial distribution of the vibrational modes inside a molecule[25,26], as well as chemical and structural changes[29,30] of a single molecule on top of a metallic surface have been studied by TERS experiments performed in a scanning tunnelling microscope (STM). However, since vibrational coherences evolve on a timescale from a few tens of femtoseconds to a few picoseconds[35], their time evolution cannot be tracked by common TERS experiments performed with continuous wave (CW) lasers. Hence the need for the imple-mentation of time-resolved TERS with the appropriate time resolution to image chemical reaction dynamics and ultrafast phase transitions in materials[36].

The realisation of time-resolved coherent Raman spectroscopy in an STM would not only allow for the observation of ultrafast dynamics in the time domain but would also provide unprecedented spatial and energy resolutions[37], thus allowing for single-molecule dynamical studies. However, despite significant efforts exploiting the plasmonic enhancement of metallic nanoantennas for CRS in molecules[38–43], achieving such a capability in an STM is still challenging and has not been realised so far. Here we demonstrate that such an approach is feasible and apply it to obtain temporal information on the motion of vibrational wave packets in single-graphene nanoribbons (GNRs).

GNRs have fascinating electronic and optical properties, which makes them ideal candidates for future quantum electronic devices[44]. Thus, understanding vibrational and electronic dynamics in GNRs at the single-molecule level is of great importance. We show that vibra-tional (phonon) coherences, i.e., the vibrational wavepacket, induced in a single-graphene nanoribbon (GNR) grown on top of a Au(111) surface can be tracked by three-pulse broadband CARS in an STM. By performing time-resolved measurements, we determine the phonon dephasing (~440 fs) and population decay times (~1.8 ps), and demonstrate that the beatings between different phonon modes associated with the vibrational wave packets can be selectively gen-erated by controlling the delay between two ultrashort pulses. Our approach reveals that the vibrational coherences of a single GNR evolve on time scales as short as ~70 fs. By analysing a two-dimensional frequency correlation spectrum, we are able to assign deterministically the quantum couplings between various phonon modes of the GNR.

## Results

### Stokes and anti-stokes Raman spectroscopy of a GNR

In our experiments, the basic signatures (frequencies and their amplitudes) of the Raman modes of a single GNR were studied by tip-enhanced Raman spectroscopy (TERS) excited both with a CW laser and with ultrashort laser pulses (Fig. 1a)[37]. An electrochemically etched Au tip was used for plasmonic enhancement of the TERS signal. 7-armchair graphene nanoribbons (7-AGNRs) on Au(111) surface were fabricated by a surface-supported bottom-up synthesis from 10,10′-dibromo-9,9′-bianthryl (DBBA) precursor molecules[45]. All measure-ments were conducted at a temperature of 90 K. An STM image of GNRs is shown in Fig. 1b. When the GNRs lie flat on the surface in the absence of the Au nanotip, we did not detect an appreciable TERS signal, due to their weak Raman scattering cross-section. Nevertheless,

a strong TERS signal appears when the Au tip is placed on top of the GNR extremity and then approached until atomic point contact forms. A pronounced enhancement of the TERS signal in molecular point contacts has been earlier reported in STM[46,47]. We note that the application of tunnelling currents of a few μA to establish atomic point contact between the GNR and the Au nanotip does not damage the GNR[48–50] (please see Fig. S2 in the SOM). In this work, all the TERS spectra were measured with the GNR in atomic point contact with the Au nanotip under constant current feedback conditions of the STM (see Supplementary Information, Section I for details).

A TERS spectrum measured with CW laser excitation is shown in Fig. 1c. The measured peaks can be assigned to the various phonon modes of the GNR. The spectral peak located around 1580 cm$^{-1}$ can be assigned to the G mode, corresponding to a carbon-carbon stretching mode, whereas the D-like modes at 1230 cm$^{-1}$ and 1330 cm$^{-1}$ are related to the edge termination of GNRs[18,51]. The general features of the Raman spectrum are well reproduced by the density functional theory (DFT) simulated spectrum shown by the purple-curve in Fig.1c (see also Supplementary Information, Section VIII). Apart from the D-like and G modes, a few additional weaker Raman peaks due to other vibrations can be observed[52], which are also present in the calculated spectrum. In the theoretical simulations, we have considered finite size 7-AGNR in slightly bent geometries to account for the formation of the atomic point contact between the GNR and the Au nanotip. The calculated Raman spectrum barely changes with the bending angle in a range from 2º to 10º and remains qualitatively the same irrespective of the GNR length (Fig. S12 and Fig. S13 in Supplementary Information). A detailed description of the theoretical calculations of the Raman active modes and associated displacements of the atoms is discussed in Section VIII of the Supplementary Information.

In the TERS spectra measured with ultrashort laser pulses, the spectral resolution is mainly determined by the spectral width of the exciting ~500 fs long laser pulses, hereafter referred to as the 'probe' pulses, which is ~30 cm$^{-1}$ for pulses centred at ~728 nm with a band-width of ~1.5 nm[37]. Such a resolution is enough for the identification of all the Raman peaks in most of the materials. Figure 1d shows the Stokes Raman spectrum measured with the probe pulses. The Raman peaks for both the D-like mode at 1330 cm$^{-1}$ and the G mode at 1580 cm$^{-1}$ can be clearly identified, with a reduced spectral resolution compared to the CW TERS. The contribution of the Raman mode at 1230 cm$^{-1}$ is also visible as a shoulder. A linear dependence of the intensity of the Stokes Raman signal with respect to the power of the incident probe pulses was measured, indicating that the TERS signal purely arises from a spontaneous Raman scattering process (see Fig. S4 in Supplementary Information).

The intensity of the anti-Stokes Raman signal depends on the populations of the higher initial vibrational levels, which can either be pre-existing or can be excited. At our experimental temperature of ~90 K, the intensity ratio between anti-Stokes and Stokes signals is ~10$^{-7}$, based on a simple estimation from the Boltzmann distribution[51]. We note that since very low laser powers (a few mW) were used in the experiments, the laser-induced thermal effects in the STM junction were negligible[53]. Thus, with CW laser excitation, negligible anti-Stokes scattering is measured from the GNR. In contrast, when exciting with laser pulses of similar incident power, significant anti-Stokes scattering was measured (Fig. 1e), indicating the excitation of phonons in the GNR. The latter is possible due to the higher peak intensity of the ultrashort laser pulses compared to that of the CW laser[51,54]. Hence, the variation of the signal strength of the anti-Stokes scattering upon increasing the power of the incident probe laser pulses shows a quadratic dependence[33]. This is a consequence of the fact that the Raman scattering cross-section is not only proportional to the laser intensity but also to the probability of exciting higher vibrational states, which also varies linearly with the laser intensity (see Fig. S4 in Supplementary Information).

## Femtosecond broadband coherent anti-stokes Raman spectroscopy

To track the temporal evolution of phonon wave packets in an individual GNR, three-pulse tip-enhanced CARS experiments were performed. The vibrations of a GNR were coherently excited by a combination of two broadband laser pulses of ~100 fs duration, noted hereafter as 'pump' and 'Stokes' pulses. The frequency difference between the pump and Stokes pulses matches the entire bandwidth of the vibrational spectrum of a GNR up to ~2400 cm$^{-1}$, thus leading to a coherent superposition of the vibration levels (a vibrational wavepacket) on their simultaneous interaction with a GNR. The coherent evolution of the phonons was traced in real-time by a delayed probe pulse, which triggers the anti-Stokes scattering from the vibrationally excited components of the generated wavepacket, as schematically shown in Fig. 2a. The delays between pump and Stokes pulses ($\tau_{12}$), and between Stokes and probe pulses ($\tau_{23}$) were precisely and independently controlled. The spectra of the pump, Stokes and probe pulses are shown in Fig. 2b (see also Supplementary Information, Section II for details of the experimental set-up).

We first analyse the characteristic features of the CARS signal measured when the delays between the three pulses are zero ($\tau_{12} = 0$, $\tau_{23} = 0$). As shown in Fig. 2c, a set of sharp anti-Stokes Raman peaks appears on top of a broad background. The three main Raman peaks of the GNR at 1230 cm$^{-1}$, 1330 cm$^{-1}$, and 1580 cm$^{-1}$ can be clearly identified. Interestingly, several Raman peaks, which are weak or even absent in the spontaneous Raman spectra in Fig. 1c–e, become pronounced in the CARS spectrum. This suggests that coherent (impulsive) excitation of the vibrational modes of the GNR by broadband pump and Stokes

pulses provides an additional enhancement of the Raman scattering cross-section on top of the plasmonic enhancement in TERS. The plasmonic emission spectrum from the nanocavity formed between the Au nanotip and the Au(111) surface (black-curve in Fig. 2c) closely resembles the anti-Stokes scattering spectrum measured on the clean Au(111) surface (blue-curve in Fig. 2c), hence, indicating that the enhancement of the Raman signal by the nanocavity plasmons is essential in tip-enhanced CARS. We note that the count number for CARS performed in a single molecule cannot be compared with the count number in bulk samples, and the signal level of 10 counts/s is sufficient for single-molecule studies[16,55].

The intensity of coherent Raman scattering is determined by a third-order nonlinear response (four-wave mixing) of the interacting system, $I^{(3)}_{CARS} \sim I_{Probe} \times (I_{Pump} \times I_{Stokes})$, where $I_{Probe}$, $I_{Pump}$, and $I_{Stokes}$ refer to the local intensity of the three individual pulses, respectively. However, the CARS signal can also originate from the contribution of only two, instead of three laser pulses. This is referred to as two-colour CARS, in which the vibrational coherence is generated by a combination of a pump (or Stokes) and a probe pulse, and subsequently detected by the same probe pulse[56]. Although two-colour CARS also results from a third-order nonlinear response, its intensity follows a significantly different dependence with respect to the intensity of the interacting pulses: $I^{(3)}_{2-CARS} \sim (I_{Probe})^2 \times I_{Pump}$ or $I^{(3)}_{2-CARS} \sim (I_{Probe})^2 \times I_{Stokes}$. To verify whether the origin of the CARS spectrum shown in Fig. 2c comes from two- or three-colour CARS, we measured the variation of the CARS spectra with respect to the total power of the pump and Stokes pulses ($I_{Pump} + I_{Stokes}$), as shown in Fig. 2d. A very different power dependence is measured for the

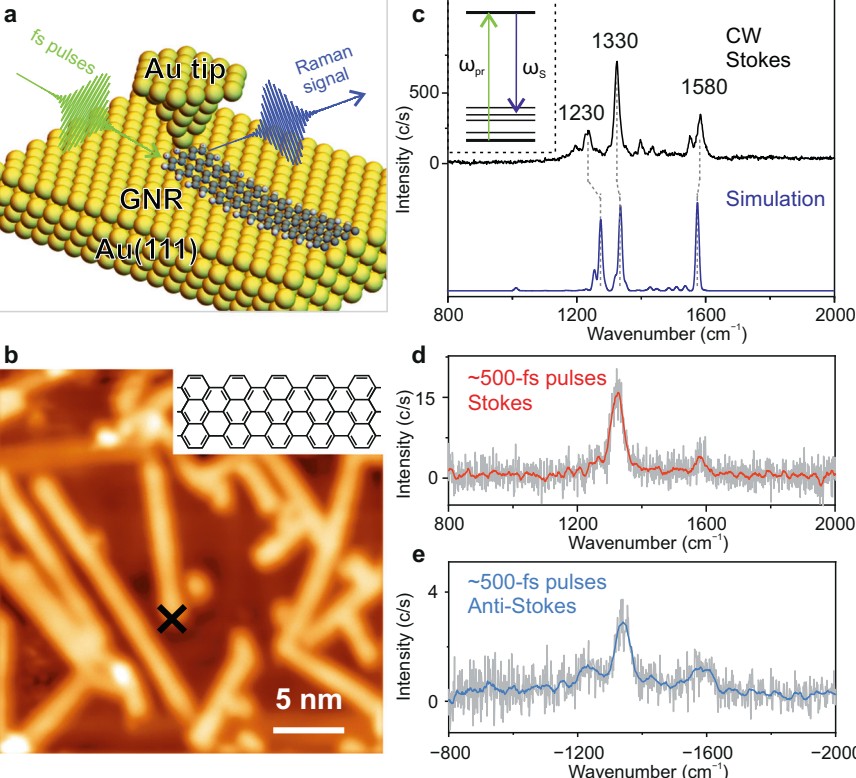

**Fig. 1 | Ultrashort pulse excited tip-enhanced Raman spectroscopy (TERS) of a graphene nanoribbon (GNR). a** Schematic illustration of TERS of a single GNR excited by ~500 fs long ultrashort laser pulses. **b** STM image of GNRs adsorbed on Au(111) measured at a bias voltage of $V = 1$ V, and a set-point current of $I = 20$ pA. Top-right panel shows the chemical structure of a GNR. **c** Inset: Schematic illustration of spontaneous (incoherent) Raman transitions excited by ultrashort laser pulses. Black-curve: TERS spectrum acquired with continuous wave (CW) laser excitation, wavelength $\lambda = 633$ nm, laser power $P = 1.0$ mW, bias voltage (V) of 10 mV, set-current ($I$) of 8 nA. Purple-curve: DFT simulated Raman spectrum for an L7 GNR (see Fig. S13 in the Supplementary Information). **d** Stokes Raman spectrum acquired with ~500 fs laser pulses ($\lambda = 728$ nm, $P = 1$ mW, $V = -500$ mV, $I = 4$ μA). **e** Anti-Stokes Raman spectrum acquired with ~500 fs long laser pulses ($\lambda = 728$ nm, $P = 0.93$ mW, $V = -500$ mV, $I = 8$ μA).

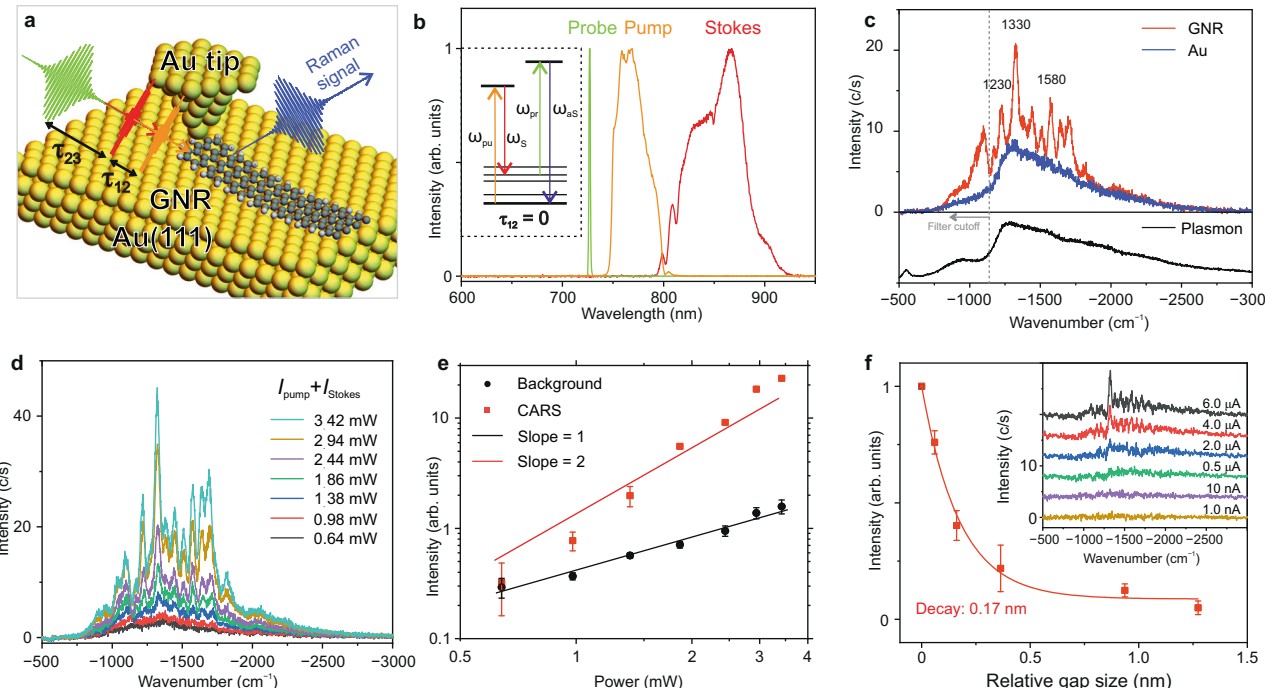

**Fig. 2 | Femtosecond broadband coherent anti-Stokes Raman spectroscopy (CARS) of a single-graphene nanoribbon. a** Schematic illustration of time-resolved CARS of a GNR. **b** Spectrum of the ultrashort pump (750–805 nm), Stokes (805–920 nm) and probe laser pulses. Pump and Stokes laser pulses are ~100 fs long, with their central wavelength being ~780 nm and ~850 nm, respectively. Probe pulses are ~500 fs long, with a bandwidth of 1.5 nm, centred at 728 nm. Inset: Schematic illustration of coherent Raman transitions stimulated by pump and Stokes pulses and tracked by probe laser pulses. **c** Comparison of the CARS spectra when the Au nanotip is placed on a clean Au(111) surface (blue-curve) and when the Au nanotip is placed on top of one of the extremities of the GNR in atomic point contact (red-curve) ($P_{pump} = 1.31$ mW, $P_{Stokes} = 1.13$ mW, $P_{probe} = 1.3$ mW). The bias voltage and tunnelling current at the STM are $V = -200$ mV and $I = 8$ µA, respectively. The pump, Stokes and probe pulses are placed in perfect temporal overlap ($\tau_{12} = 0$, $\tau_{23} = 0$). The plasmonic emission spectrum of the nanocavity of the STM

(Au nanotip and Au(111) surface) is shown by the black-curve ($V = 3$ V, $I = 10$ nA). **d** Series of CARS spectra measured as a function of the power of the pump and Stokes pulses while the power of the probe pulse remains unchanged. The power ratio between the pump and Stokes pulses is ~1. **e** Scaling of the integrated CARS signal (red-curve) and the broadband background (black-curve) with increasing power of the pump and Stokes laser pulses plotted in a dual logarithmic-plot. The intensities of the CARS signal and the background were evaluated by integrating the area under the Raman peaks at $-1330$ cm$^{-1}$ and in the broad (featureless) spectral range from $-2500$ cm$^{-1}$ to $-2300$ cm$^{-1}$, respectively. **f** Variation of the integrated intensity of the CARS signal with the relative plasmonic nanocavity size ($P_{pump} = 1.4$ mW, $P_{Stokes} = 1.2$ mW, $P_{probe} = 1.3$ mW, $V = -200$ mV). The inset shows the CARS spectra as a function of increasing tunnelling current (decreasing plasmonic gap size). Error bars in **e** and **f** show the standard deviation.

Raman peaks and for the background, as shown in Fig. 2e, indicating very different signal generation mechanisms. The quadratic dependence of the intensity of the Raman peaks with respect to $I_{Pump} + I_{Stokes}$ shows that the coherent Raman peaks are indeed generated via a three-colour CARS mechanism. The linear dependence of the background intensity with respect to $I_{Pump} + I_{Stokes}$ excludes the contribution of non-resonant background (NRB) resulting from the coherent interference between the three pulses as its origin. The broad background primarily arises from the spontaneous anti-Stokes scattering from electrons out of equilibrium (hot electrons) that are locally excited by the ultrashort pulses, which is an incoherent process.

We further explore the extent of the spatial localisation of the tip-enhanced CARS signal below the Au nanotip. The strong interaction of ultrashort laser pulses with the GNR and the plasmonic nanocavity formed between the Au nanotip and the Au(111) surface leads to a strong dependence of the CARS signal with respect to the nanocavity size. In our experiments, since the GNR is attached to the nanotip, a relative change of the tip height results in the lifting of the GNR from the Au(111) surface plane. As shown in Fig. 2f, the variation of the intensity of the CARS signal upon increasing the nanocavity (gap) size, $d$, can be fitted with an exponential function, $I_{CARS} \propto \exp(-d/k)$, with a decay length of $k = 170$ pm, which suggests a sub-nanometre spatial resolution of the CARS signal[19,23]. It is worth mentioning that the positions of

the Raman peaks in the CARS spectra remain mostly unchanged despite variations in the nanotip height. This observation is in agreement with DFT simulations of the Raman spectra performed for different bending angles of the GNR.

## Dephasing dynamics of coherent phonons in a GNR

Dephasing dynamics of coherently excited phonons can be traced in real-time by time-resolved CARS[39,57]. Here, we overlap the pump and Stokes pulses in time ($\tau_{12} = 0$ fs) to maximise the impulsive excitation of the vibrational coherence in a single GNR. The variation of the CARS signal as a function of the delay between the probe pulse and the pump and Stokes pulses ($\tau_{23}$) is shown in Fig. 3a. No CARS signal is measured when the probe pulses arrive before the pump and Stokes pulses ($\tau_{23} \ll 0$), as the vibrational coherence has not been excited yet. The intensity of the CARS signal reaches its maximum when the three pulses overlap in time ($\tau_{23} = 0$) and gradually decreases when the delay $\tau_{23}$ increases. Figure 3b shows three selected spectra at delay times $\tau_{23}$ of $-0.5$ ps, 0 ps and 0.5 ps, depicting the increase, maximum, and decay of the CARS signal, respectively. The integrated Raman intensity as a function of the delay $\tau_{23}$ from the measurements in Fig. 3a is depicted in Fig. 3c. We observe a rising time of ~260 fs, which is in accordance with the duration of the probe pulses (~500 fs). A dephasing time ($T_2/2$) of ~440 ± 70 fs can be retrieved by an exponential fit of the experimentally measured points along the positive delay axis, as shown by the black-curve in Fig. 3c.

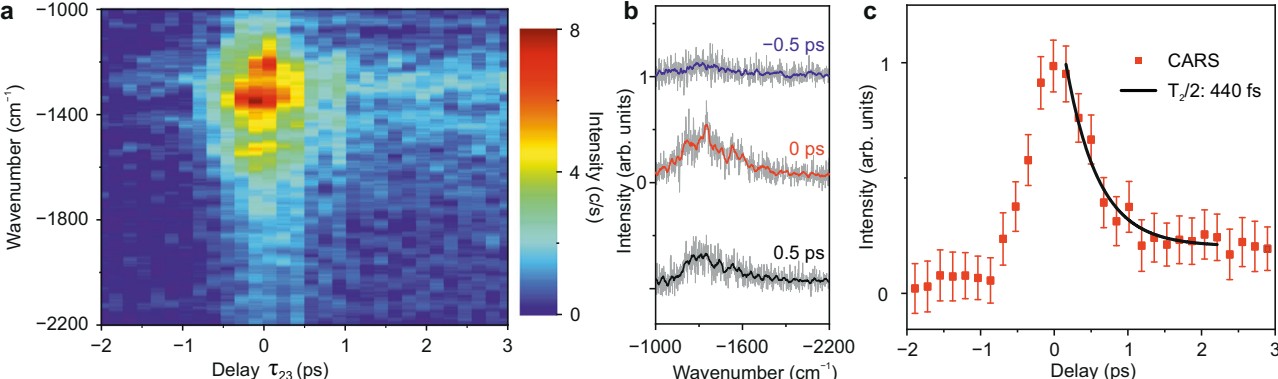

**Fig. 3 | Dephasing dynamics of impulsively excited phonons. a** False colour representation of time-resolved CARS spectra of a single GNR acquired at various delays ($\tau_{23}$) of the probe pulse with respect to the pump and Stokes pulses. Measuring parameters: $P_{pump} = 1.4$ mW, $P_{Stokes} = 1.2$ mW, $P_{probe} = 1.3$ mW, $V = -50$ mV, $I = 8$ μA, acquisition time of 10 s per spectrum. The pump and Stokes pulses overlap in time ($\tau_{12} = 0$). **b** CARS spectra at three $\tau_{23}$ relative delays between the probe and pump/Stokes laser pulses from the measurement in **a**, the relative delays are annotated on top of each spectrum. **c** Integration of the Raman peaks from the individual spectral measurements in **a** as a function of the relative delay ($\tau_{23}$) between the probe and pump/Stokes laser pulses (red points). Positive delay refers to the situation when the probe pulse arrives after the pump and Stokes pulses. An exponential fitting of the measurement reveals a dephasing time ($T_2/2$) of the impulsively excited phonons of ~440 fs. Error bars show the standard deviation.

The dephasing time $T_2$ determined in this way is a total dephasing time, which contains contributions from the pure dephasing time $T_2^*$ and an apparent dephasing time $T_1$ due to the spontaneous decay of the excited vibrational states into the ground vibrational state (hereafter called population decay time), $2/T_2 = 2/T_2^* + 1/T_1$. By performing time-resolved incoherent anti-Stokes Raman spectroscopy (IARS) (see Fig. S5 in Supplementary Information), a population decay time ($T_1$) of ~1.8 ps was measured. Therefore, $T_2 \ll T_1$ and, consequently, $T_2^* \approx T_2$, so that, in practice, the measured dephasing time is a pure dephasing time. Our reported value is close to those obtained for other carbon-based materials, such as carbon nanotubes[57] and graphene[58], using time-resolved CARS. It is also worth stressing that the background in the CARS spectra (Fig. 3a) decays much more slowly than the Raman peaks from the GNR. This further corroborates that the broadband background in the CARS spectra comes from an incoherent process, as opposed to the non-resonant background, which would only exist when the three pulses overlap in time.

**Tracking and controlling coherent phonon wave packets in a GNR**

The interaction of two broadband ultrashort laser pulses (pump and Stokes) with an individual GNR leads to an impulsive nonlinear excitation of vibrational coherences, i.e., it generates a coherent vibrational wavepacket (see Fig. 4a). As mentioned above, the frequency difference between the two pulses matches the energy separation between all the vibrational levels in the GNR up to ~2400 cm⁻¹. Here, we probe ultrafast quantum beatings between vibrational states involved in this coherent superposition by selective preparation of vibrational wave packets. The initial phase and population of the vibrational states entering this coherent superposition are modulated by changing the delay between the pump and Stokes pulses[59], which leads to periodic variations in the anti-Stokes scattering induced by the probe pulse from the different vibrational states (see the model described in section IX of the Supplementary Information for more details). These periodic variations are superimposed to an exponentially decreasing background due to dephasing (see above) and are convoluted with the duration of the probe pulse[60], but all in all they allow us to uncover the quantum beatings inherent to the generated vibrational wavepacket.

In these measurements, the probe pulse temporally overlaps with the Stokes pulse ($\tau_{23} = 0$), so that only the relative delay between the pump and Stokes pulses (or equivalently, between the pump and probe pulses), $\tau_{12}$, is varied. A series of anti-Stokes spectra measured as

a function of $\tau_{12}$ is shown in Fig. 4b. One can observe clear oscillations in the spectral intensity of the D-like and G modes at 1230 cm⁻¹ and 1580 cm⁻¹ indicated by coloured arrows in Fig. 4b. The variation of the spectral intensity of these two Raman modes as a function of the $\tau_{12}$ delay is shown in Fig. 4c. To understand the origin of the oscillations of the spectral intensity, we Fourier transform the temporal evolution at each wavenumber in Fig. 4b (horizontal cuts) to produce a two-dimensional (2D) map of quantum beats. Figure 4e shows a false colour representation of the intensity of the Fourier transformations as a function of the beating frequencies (x-axis) and the frequency of the measured anti-Stokes scattering (y-axis). A crosscut of the 2D map in Fig. 4e at the frequency of the phonon modes at 1230 cm⁻¹ and 1580 cm⁻¹ is shown in Fig. 4f. Identical beating frequencies of ~350 cm⁻¹ are obtained for both Raman modes, which matches quite well the energy difference between the two involved vibrational levels. The coherence between these two phonon modes leads to an out of phase oscillation of the corresponding anti-Stokes scattering upon variation of the delay between the pump and Stokes pulses, as shown in Fig. 4c. Similarly, the beating frequency at ~500 cm⁻¹, represented by a white solid circle in Fig. 4e, arises from the quantum beating between the phonon modes at 1230 cm⁻¹ and around 1700 cm⁻¹. The strong beating frequency at ~190 cm⁻¹ for the mode at 1230 cm⁻¹ possibly comes from the interference with the low frequency phonon mode at around 1050 cm⁻¹. In our approach, the time resolution provided by the probe pulses is not as critical as in a conventional CARS to track the quantum oscillations (beatings) between various phonon modes in the system. The observed oscillations in the anti-Stokes scattering (Fig. 4c) can be modelled by using second-order time-dependent perturbation theory together with an impulsive approximation to describe the excitation of the vibrational levels of the GNR when interacting with delay-controlled pump and Stokes pulses (see section IX in Supplementary Information).

Since the GNR has multiple vibrational levels, the analysis of the 2D quantum beat map (Fig. 4e) is complex due to the possibility of several couplings (oscillations) involving the same phonon levels. To further confirm that our experimental approach is capable of tracking vibrational coherences, we performed experiments in ambient conditions on single-walled carbon nanotubes (CNTs), which have fewer phonon levels by using identical pulse parameters (see section V in Supplementary Information). An oscillation of the anti-Stokes scattering from different vibrational levels of the CNTs was measured at the frequency corresponding to the energy differences between those

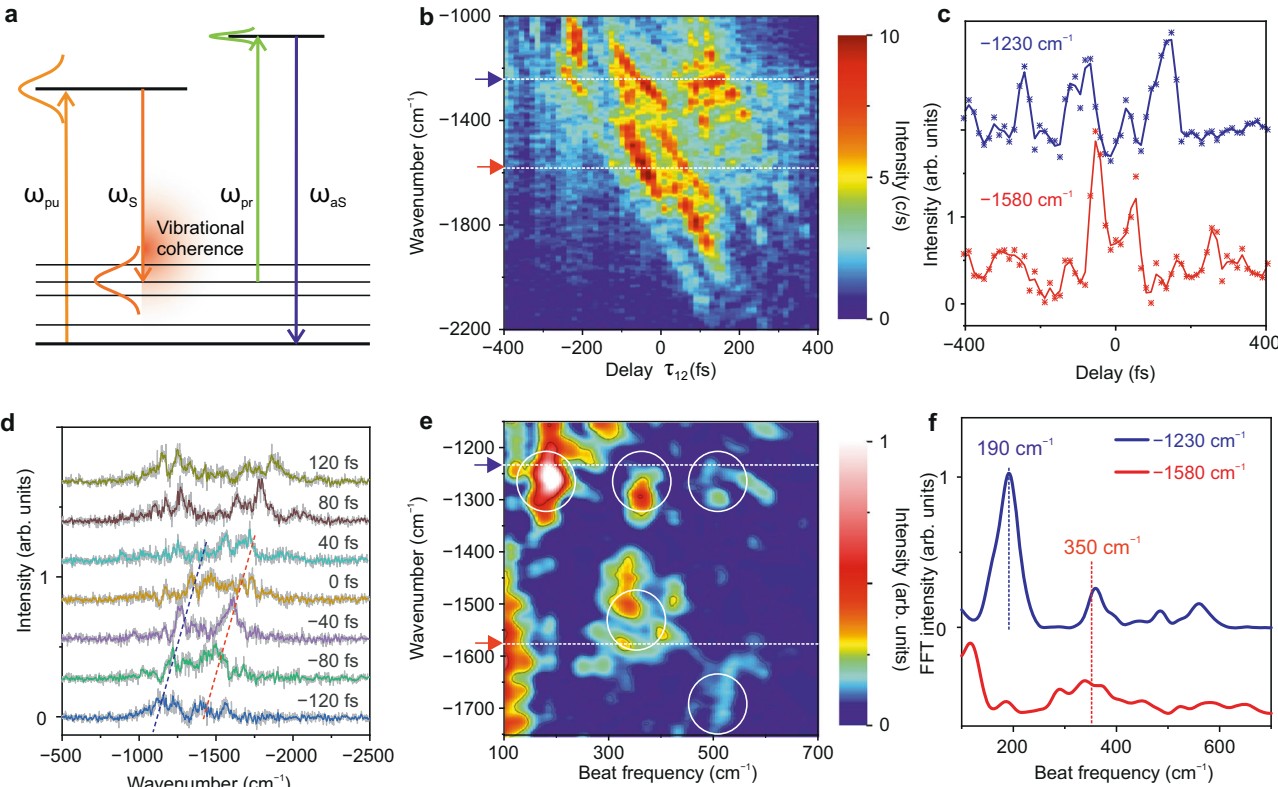

**Fig. 4 | Tracking Ultrafast Coherent Phonon Oscillations in a single GNR.**
**a** Schematic illustration of impulsive excitation of a vibrational wavepacket in a GNR interacting with ultrashort broadband pump ($\omega_{pu}$) and Stokes ($\omega_S$) pulses. The phase modulation of the coherent vibrational wavepacket launched by the delay-controlled (varying) pump and Stokes pulses is investigated by the anti-Stokes scattering ($\omega_{aS}$) triggered by the probe ($\omega_{pr}$) pulses; here $\tau_{23} = 0$ and $\tau_{12} \neq 0$. **b** False colour representation of anti-Stokes spectra as a function of the delay of the pump pulse with respect to the probe and Stokes pulses ($\tau_{12}$ modulation). Measuring parameters: $P_{pump} = 1.4$ mW, $P_{Stokes} = 1.2$ mW, $P_{probe} = 1.3$ mW, $V = -100$ mV, $I = 8$ μA, acquisition time per spectrum $t = 10$ s. The probe and Stokes pulses overlap in time ($\tau_{23} = 0$). **c** Temporal variation of the spectral intensity of two different phonon

modes at -1230 cm$^{-1}$ and -1580 cm$^{-1}$ as a function of the delay between pump and Stokes pulses (modulation of $\tau_{12}$, $\tau_{23} = 0$) from the measurement shown in **b**. The position of the two phonon modes is indicated by respective colour-coded arrows in **b**. **d** Vertically shifted representative CARS spectra from the measurement in **b**, the relative $\tau_{12}$ delays are annotated on top of each spectrum. **e** Two-dimensional map of quantum beatings of phonon modes obtained by Fourier transforming each horizontal slice of the map in **b**. The amplitude of the Fourier transformations (false colour) is plotted as a function of the beating frequencies ($x$-axis) and the measured spectral axis of the anti-Stokes scattering ($y$-axis). **f** Cross-cuts of the two-dimensional quantum beat map in **e** at the phonon modes frequencies of -1230 cm$^{-1}$ and -1580 cm$^{-1}$.

levels, as also evident from the analysis of the corresponding 2D quantum beat map (Fig. S7 in Supplementary Information). We also performed a control experiment to elucidate how pulse chirping impacts quantum beating, in which we track vibrational coherences in a graphite sample by systematic variation of the linear chirp of the ultrashort pump and Stokes pulses (see section VI in Supplementary Information). These experiments further support the proposed approach for tracking quantum coherences between excited vibrational levels.

The vibrational peaks of the anti-Stokes scattering shown in Fig. 4b also undergo a spectral shift on top of the intensity oscillations. All the peaks shift when varying the $\tau_{12}$ delay, as indicated by the tilted dashed lines for the phonon modes at -1230 cm$^{-1}$ and -1580 cm$^{-1}$ in Fig. 4d. The origin of this spectral tilt is related to the positive chirp of the pump and Stokes pulses. Although both pulses are compressed at the laser source, they become positively chirped when they reach the STM nanotip, having a duration of -100 fs. Second-harmonic frequency-resolved optical gating (FROG) measurements were conducted to characterise the temporal profile of the laser pulses (see section VII in Supplementary Information). A simple relationship accounting for the time evolution of the frequency of the pump and Stokes pulses can be expressed as $\omega_{Stokes}(t) = \omega_{SO} + \alpha t$ and $\omega_{Pump}(t) = \omega_{PO} + \beta t$, where $\alpha$ and $\beta$ represent the positive chirp of Stokes and pump pulses, respectively. Since the chirp of Stokes and

pump pulses is similar, we can assume $\alpha \approx \beta$. A variation of the delay ($\tau$) between the two pulses modulates the frequency of the coherent oscillation: $\omega_{Pump}(t+\tau) - \omega_{Stokes}(t) = (\omega_{PO} - \omega_{SO}) + \alpha\tau$. This shift of the coherent excitation frequency modifies the position of the Raman peaks of the GNR by -150 cm$^{-1}$, in close analogy to a driven classical harmonic oscillator, where the oscillator can still be excited (and probed) by an off-resonant excitation frequency. Nevertheless, this frequency cannot be too far from the resonance frequency, else there would be no excitation anymore. At a certain delay $\tau$ between pump and Stokes pulses, when the coherent excitation frequency matches the energy gap (resonant excitation) of a particular vibrational level, its spectral intensity will be higher compared to the off-resonant case, as shown in Fig. 4b and Fig. 4d. A similar spectral shift in the Raman peaks has also been measured in CNTs as shown in Fig. S7 in the Supplementary Information. This demonstrates that coherent phase control in CARS can be achieved by the implementation of delay-controlled positively chirped pump and Stokes pulses, which manifests in the spectral shift of the Raman peaks. In the presence of competing pathways in the four-wave mixing process, the contribution of the coherent Raman signal can be controlled with respect to the non-resonant background[61].

The capability to directly (and locally) capture dephasing dynamics as well as quantum oscillations between the vibrational levels of a single GNR opens new avenues to coherently prepare and

manipulate vibrational wave packets in individual molecules and quantum materials, so as to drive them to a preferred pathway in light-induced transformations. Our work paves the way to study the intricate role of the vibrational degrees of freedom of a molecule/material undergoing a geometrical/chemical transformation or a photo-induced charge-transfer process in real-space and real-time, with sub-nm (spatial), ~50 fs (temporal) and ~meV (energy) resolutions, simultaneously. This is an important step towards the spatio-temporal control of the properties of quantum materials.

## Methods

### Sample and tip preparation

The experiments were performed in a custom-built scanning tunnelling microscope (STM) operating in ultra-high vacuum conditions (~$5 \times 10^{-10}$ mbar), and at liquid nitrogen temperature (~90 K). Au(111) surfaces were prepared by repeated cycles of sputtering with 1.0 keV $Ar^+$ ions and thermal annealing at ~500 °C. To fabricate graphene nanoribbons, 0.5 ML of 10,10'-dibromo-9,9'-bianthryl molecules (DBBA) were sublimated on top of the clean Au(111) sample held at room temperature. 7-armchair graphene nanoribbons (7-AGNRs) were obtained by post-annealing the sample at 400 °C for 15 min. Electro-chemically etched Au tips were used in all the experiments to enhance the electric field confinement of the laser pulses. All topographic images presented in the current work were acquired in the 'constant current mode' of the STM.

All TERS spectra from the GNRs presented in the current work were measured from a single GNR in atomic point contact with the Au nanotip. To establish the atomic point contact, the Au nanotip was placed on top of the GNR extremity and the tip-sample distance was reduced by increasing the tunnelling current to ~6 µA in the constant current mode.

### Optical setup

The ultrafast laser system used in the current work is a Ti:Sapphire oscillator (Element™ 2, Newport Spectra-Physics) which produces laser pulses of ~6 fs duration with a bandwidth spanning from 650 nm to 1050 nm at a repetition rate of ~80 MHz. Probe pulses centred at ~728 nm with a duration of ~500 fs were generated by narrowband filtering (Ultra Narrow Bandpass Filter 728.1/1.5, AHF) of the broad-band ~6 fs long laser pulses. Pump pulses (~750–805 nm) and Stokes pulses (~805–920 nm) were also generated by bandpass filtering of the broadband laser pulses. Two precise (resolution ~0.1 µm) delay stages were used to control the delay time $\tau_{12}$ between the pump and Stokes pulses and $\tau_{23}$ between the Stokes and probe pulses. An achromatic lens (diameter: 50 mm; focusing length: 75 mm) was mounted inside the UHV chamber to focus the laser beams onto the apex of the Au tip. The TERS signal was collected through the same achromatic lens and then focused onto the entrance slit of a spectrometer (Kymera 328i, ANDOR) and detected by a thermoelectrically cooled charge coupled device (iDus 416, ANDOR). TERS experiments were also performed with CW excitation by using a Helium-Neon (He-Ne) CW laser (HNL150L, Thorlabs) centred at ~633 nm.

## Data availability

The data that support the findings of this study are available from the corresponding author on request.

## Code availability

The codes used for the DFT simulations in this study are available from the corresponding author on request.

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

## Acknowledgements

We thank Wolfgang Stiepany and Marko Memmler for the technical support. Work supported by the European COST Action AttoChem. All calculations were performed at the Mare Nostrum Super-computer (BSC-RES) and the Xula Supercomputer (CIEMAT-RES) of the Red Española de Supercomputación (RES), and the Centro de Computación Científica de la Universidad Autónoma de Madrid (CCC-UAM). FM acknowledges support by the MICINN projects PID2019-105458RB-I00, the "Severo Ochoa" Programme for Centres of Excellence in R&D (CEX2020- 001039-S), the "María de Maeztu" Programme for Units of Excellence in R&D (CEX2018-000805-M) and the Comunidad de Madrid Synergy Grant FULMA-TEN. A.M.J acknowledges the Alexander von Humboldt Foundation for financial support.

## Author contributions

M.G. conceived the project and designed the experiments. K.K. super-vised the project. Y.L., M.G. and A.M.J. built the experimental set-up, performed the experiments and analysed the experimental data. M.P. and F.M. developed the second-order perturbation theoretical model and performed the DFT calculations. All authors interpreted the results and contributed to the preparation of the manuscript.

## Funding

## Competing interests

The authors declare no competing interests.
