## [Peer Review File · Nature Communications]

Reviewers' Comments:

Reviewer #1:

Remarks to the Author:

I had previously reviewed this manuscript for Nature Materials and recommended its transfer to Nature Communications. Having read the replies of the authors to reviewer 4, and in light of the recently published paper Science 379, 1049–1053 (2023) which uses electroluminescence in an STM and shows Raman spectra in good agreement with the CARS spectra reported in the present paper, I can confirm my recommendation to publish this work in Nature Communications.

Reviewer #2:

None

Response to the comments of Reviewer #4

The responses provided by the authors to points raised by reviewer 4 are not satisfactory. I do not recommend publication without addressing the following:

1) Regarding Raman vs CARS, given by the third-order polarization and its square, it behooves the authors to be specific about what is learned in one that is not present in the other - the general claims made in the rebuttal are incorrect.

We have clarified this point in the previous peer review process. In the current work, we have performed time-resolved CARS to track the quantum coherences between various vibrational levels of a single GNR in real-time. Real-time tracking of quantum coherences is the key to understanding various phenomena related to charge/energy transfer in photosynthetic systems, solar cells, etc.. This information simply CANNOT be obtained by a static measurement (CW excitation). Indeed, the estimation of the vibrational lifetime is possible by analysing the linewidths of spontaneous Raman emission, which nevertheless, can be done much more meticulously (without errors) in the time-domain measurements, as we also argued previously. Nevertheless, the current work clearly goes much beyond that, by employing CARS we tracked vibrational coherences between the phonon modes of a single GNR.

2) "Imaging" has a simple meaning in English. Its misuse is not acceptable once pointed out. "Observing" would be appropriate here. The objection is not semantic. The authors want to suggest "simultaneously achieve atomic temporal and spatial resolution (femtoseconds and sub-nm)" - there simply is no spatial data, therefore no spatial resolution to talk about. The vertical approach simply means signal at effective contact – but no image (x,y-dependent signal).

In the previous two review rounds, we had explained our reasoning behind using the word 'imaging' in the title of our manuscript. A well-characterized quantum system, in which the energy of all its quantum levels (eigen states) can be experimentally determined, and the time evolution of all the involved eigen states and their couplings with each other can be measured experimentally, then it is possible to reconstruct the motion (externally stimulated) of the quantum system under investigation. This approach of reconstructing electronic or atomic motion in molecules has been broadly referred to as *imaging*, *observing* or *seeing* in many papers. Please see, e.g. *Nature Photonics* **8**, 650-656 (2014); *Nature* **432** (7019), 867-871 (2004); *Nature* **466**, 739–743 (2010).

In our experiments, the area of the GNR, where we excite and probe phonon dynamics, is precisely known. The CARS signal is obtained from a determined single 7-AGNR. **The vertical decay length of ~ 1.7 angstroms of the CARS signal as shown in Fig. 2f (main-text) connotes to the atomic scale localization of the CARS signal. This implies a very high spatial resolution (sub-nm) of the CARS signal both in the vertical (z) as well as in the lateral (x, y) dimensions.** This experimental evidence suggests that the time-resolved CARS as implemented in our work has the capability to '*simultaneously achieve femtosecond and sub-nm*'

resolutions, and can be applied to capture atomic motions in single molecules and novel materials.

3) Indeed, the EL spectra do suggest that GNR is being probed, at the same time, they suggest that spectra presented as CARS are not. EL is incoherent emission. It behooves the authors to recognize this and present the spectra as such.

We understand that electroluminescence (EL) is an incoherent process in contrast to CARS, which is a coherent process. The EL spectra as reported in the Science article (Science **379**, 1049–1053 (2023)) shows the vibrational peaks that are in very good agreement with the Raman peaks reported by us. This provides additional evidence that the measured vibrational modes in the CARS spectrum indeed originate from the GNRs, as also commented by Reviewer #1. The purpose of augmenting our reasoning by showing the work of another group was to transparently show that the **GNR is not damaged in our measurements and the CARS spectra presented by us is from an intact GNR** and not from a broken GNR (adventitious carbon) as imagined by the reviewer.

4) A more convincing argument is made in the response that the GNR is not destroyed during the measurement. Showing that the signal disappears after dropping the GNR would be more to the point.

We are happy to see that the reviewer is now finally convinced that **GNR is not damaged in the measurements and that the CARS spectra reported in the current work do not arise from a broken GNR or adventitious carbon as previously referred to by the reviewer**. In our experiment, I-z curves were measured frequently during the CARS measurements to confirm that the STM tip remains in atomic point contact with the GNR. **The behaviour of the I-z curve would be completely different if the GNR detaches from the tip**. We have also clearly written in the main text, page 4: *“In this work, all the TERS spectra were measured with the GNR in atomic point contact with the Au nanotip under constant current feedback conditions of the STM”*.